# Comparing Four Red/Green-Leafed Vegetables Reveals the Complementary Photoprotective Roles of Anthocyanin Accumulation and Chlorophyllase

**DOI:** 10.3390/plants14192950

**Published:** 2025-09-23

**Authors:** Ying Chen, Ruihao Zhong, Kenan Zhang, Tianyi Li, Yanan Tian, Zhaoqi Zhang, Xuequn Pang, Xuemei Huang

**Affiliations:** 1College of Life Sciences, South China Agricultural University, Guangzhou 510642, China; chenying3903@163.com (Y.C.); rhzhong@scau.edu.cn (R.Z.); xqpang@scau.edu.cn (X.P.); 2College of Horticulture, South China Agricultural University, Guangzhou 510642, China; kolanz@163.com (K.Z.); zqzhang@scau.edu.cn (Z.Z.); 3Shanghai Key Laboratory of Plant Molecular Sciences, College of Life Sciences, Shanghai Normal University, Shanghai 200234, China; aa36880@163.com (T.L.); yanantian@shnu.edu.cn (Y.T.)

**Keywords:** chlorophyllase, anthocyanin, photoprotection, leafy vegetables, red-leafed variety, green-leafed variety

## Abstract

The photoprotective role of anthocyanins in leaves is debated, as some anthocyanin-rich red leaves do not exhibit greater tolerance to high-light conditions than their anthocyanin-deficient green counterparts. In this study, we studied four leafy vegetables with both red- and green-leafed varieties: Bok Choy and Choy Sum from *Brassica rapa*, and Ramosa and Asparagus lettuce from *Lactuca sativa*. Under normal-light conditions, the red cultivars accumulated anthocyanins, the green ones did not, and all presented no photoinhibition. However, the green-leafed varieties exhibited 3–5-fold higher chlorophyllase (CLH) activity than their red counterparts. Under high-light conditions, more anthocyanins were accumulated in the red cultivars, but again, none accumulated in the green cultivars; the green cultivars showed greater CLH activity than their red counterparts. Bok Choy and Choy Sum demonstrated comparable photoinhibition between their red and green counterparts, with a similar reduction in photosynthetic activity, *F_v_*/*F_m_*, ETR, and NPQ; red Ramosa and Asparagus lettuce exhibited worse high-light tolerance than their green counterparts, with greater reductions in *F_v_*/*F_m_* and ETR. In Arabidopsis, the anthocyanin-deficient mutant *tt3tt4* (green) also induced higher *AtCLH1/2* expression than the wild-type and constitutive anthocyanin accumulation line PAP1-D (red); the *AtCLH1* overexpressor and the *clh1-1/2-2* mutant accumulated less and more anthocyanin than the wild-type, respectively. The findings suggest that CLH induction may compensate for absent anthocyanin photoprotection in green cultivars and that the two strategies may play complementary roles in photoprotection.

## 1. Introduction

Anthocyanins are the largest group of plant pigments, responsible for the red, violet, and blue colors seen in fruit, vegetative, and flower tissues. Accumulated pigments in plants can serve as both signals for pollinators and seed dispersers in flowers and fruits and as protectors of photosynthetic tissues against light damage [1,2]. Dietary consumption of anthocyanins has been associated with protection against a wide range of human diseases due to their varied bioactivity, such as anti-oxidant, anti-bacterial, and anti-cancer effects [3,4,5]. Due to the importance of these pigments in both plant and human health, their biosynthesis and functions have been subject to extensive study.

Anthocyanins are soluble flavonoid pigments that accumulate in cell vacuoles; they accumulate in the leaf epidermal cells of many plant species at certain developmental stages or when facing environmental stresses [6]. In young leaves, the photosynthetic apparatus is not fully functional [7], and harvested light energy may significantly exceed their low photosynthetic capacity and damage their apparatus [8]. Therefore, pigment accumulation in young leaves may be correlated with their higher photoprotection requirements. Under high-light conditions, plants usually induce anthocyanin synthesis, in which the pigments are believed to serve as “sunscreens” that protect their leaves from light damage [9,10]. In addition, suboptimal temperature or water conditions for plants result in decreased photosynthetic capacity and are usually accompanied by anthocyanin accumulation [11,12]. Phosphorus (P) and nitrogen (N) deficiencies result in growth reduction, carbohydrate accumulation, sugar repression of photosynthesis, and increased susceptibility to photostresses, which also induce anthocyanin accumulation [13]. Accordingly, anthocyanin accumulation has long been considered to protect against light in plants facing direct or indirect photodamage [14,15]. However, this role has recently been called into question due to findings that the physiological responses of both red and green leaves to high light were either found to be comparable, or that red leaves performed worse than their green counterparts [10].

Other than the accumulation of anthocyanins, plants have evolved multiple mechanisms to reduce photodamage, including light avoidance-associated movement of leaves and chloroplasts, light reflectance through pubescence, salt deposition, epicuticular wax layers, scavenging of reactive oxygen species (ROS), and modification of numerical photosynthetic pathways [16]. Among these strategies, the modification of photosynthetic pathways is a relatively direct photoprotective response, including reducing light absorbance via decreasing light-absorbing photosynthetic pigments, dissipating absorbed light energy as thermal energy (NPQ) in PSII, ensuring cyclic electron flow around the reaction centers of PSI and the photorespiratory pathway, etc. [17]. Among the photosynthetic apparatuses of chloroplasts, photosystem II (PSII) is the most vulnerable to light stresses. When PSII suffers photodamage, plant cells deploy a complex repair system that involves the disassembly and reassembly of the thylakoid membrane protein complex [18,19,20].

Chlorophylls (Chls) are mainly arranged in the light-harvesting complexes (LHCs) of photosystems II (PSII) and I (PSI) at the chloroplast thylakoid membranes, and play central roles in harvesting and transferring light to PSI/II reaction centers where the energy is transformed to chemical energy [21,22,23]. When not optimally regulated, they can be a “double-edged sword” for plant cells, as excess Chls may cause light-induced stress [24]. Chl catabolism must therefore be tightly regulated in response to environmental and developmental cues during the plant life cycle [25].

Chlorophyllase (CLH) was long thought to be the first enzyme in the Chl catabolic pathway and the dephytylation enzyme responsible for initiating Chl catabolism [26,27]. However, accumulating evidence demonstrates that CLH is not the enzyme active during leaf senescence; pheophytin pheophorbide hydrolase (PPH) is the key Chl dephytylation enzyme in this major catabolism process [28,29,30]. In contrast, CLHs show high expression and activity in young leaves, but not in mature or senescent leaves [31,32]; however, their function in young leaves is still unclear. Recently, we found that Arabidopsis CLH1 is associated with PSII-dismantling complexes and responsible for Chl dephytylation, facilitating core protein D1 degradation in PSII repair, and accordingly plays an important role in photoprotection [33]. However, the photoprotective roles of CLHs in other plants, and their relationships with other photoprotection processes, are unclear.

Due to the benefits that anthocyanins can confer on human health, red-leaf horticultural crops (with accumulated anthocyanins) are generally selected as leafy vegetables for consumption. Accordingly, in many leafy vegetable species, some varieties or cultivars within the same species develop red leaves, while others are deficient in anthocyanin (maintaining green leaves). In the present study, we selected four leafy vegetable species, each with both red- and green-leafed varieties, and compared their responses to various light intensities so as to specifically understand the relationship between anthocyanin accumulation and CLH activation in photoprotection. This relationship was further proven by the relevant Arabidopsis mutants or overexpressors. Our findings demonstrate that in the young leaves of plants, CLHs show extreme activation in response to high-light conditions when anthocyanin is deficient, and that CLH induction and anthocyanin accumulation may therefore represent two complementary photoprotection strategies.

## 2. Results

### 2.1. Anthocyanin Accumulation and CLH Activity in the Red- and Green-Leafed Seedlings of Four Leafy Vegetable Species Grown Under Normal Light

We selected two Brassica and two Lactuca leafy vegetable species, Bok Choy (*Brassica rapa* var. chinensis), Choy Sum (*Brassica rapa* var. Parachinensis), Ramosa lettuce (*Lactuca sativa* var. ramosa Hort; Ramo-lettuce), and Asparagus lettuce (*Lactuca sativa* var. asparagine; Aspa-lettuce), each with both red and green cultivars, to compare their responses under different light intensities. The seeds of the eight cultivars germinated and grew into seedlings with 4–5 true leaves under normal light (NL, 280 µmol·m^−2^·s^−1^) (Figure 1A). In Bok Choy and Aspa-lettuce, the red cultivars showed obvious anthocyanin accumulation, with anthocyanin contents of 0.1205 ± 0.0040 and 0.0135 ± 0.0035 mg·g^−1^ FW, respectively. In the red cultivars of Choy Sum and Ramo-lettuce, anthocyanins mainly accumulated in the leaf veins, with contents of 0.0100 ± 0.0015 and 0.0033 ± 0.0004 mg·g^−1^ FW, respectively (Figure 1B). No anthocyanins were detected in any of the green cultivars of the four species. Compared to the large difference in anthocyanin content between the red and green cultivars, fewer differences in chlorophyll content were found. No significant difference in chlorophyll was found between the red and green cultivars of the Brassica cultivars (around 6 mg·g^−1^ FW); red Aspa-lettuce contained 25% less chlorophyll than its green counterpart, while red Ramo-lettuce showed 41% higher chlorophyll than its green counterpart (Figure 1C). Under NL, no significant differences in the maximum photochemical efficiency of PSII (*F*_v_/*F*_m_) were detected among the eight vegetable varieties, with values around 0.8 (Figure 1E).

We detected CLH activity in the leaves of the eight cultivars and found that it was significantly higher in the green cultivars than in their red counterparts. The CLH activity in the green cultivars was within the range of 2–3 μmol Chlide/h/g FW, about 3–5 times higher than that in the red cultivars (Figure 1D).

These results indicate that under NL, the red cultivars of the four species accumulated anthocyanins, while the green cultivars did not. However, the green cultivars exhibited significantly higher CLH activity than their red counterparts. Under NL, neither the red nor green cultivars showed photoinhibition, suggesting that without protection from anthocyanins, the green cultivars showed no more photoinhibition/damage than the red ones, implying that they may use this high CLH activity to ensure light protection.

### 2.2. Anthocyanin Accumulation and CLH Activity in the Red- and Green-Leafed Seedlings of Four Leafy Vegetable Species Under Low Light

To observe the phenotypes of the red and green cultivars of all four vegetable species under low-light conditions (20 µmol·m^−2^·s^−1^, LL), we grew the seedlings under an LL intensity of 20 µmol·m^−2^·s^−1^. Among all eight cultivars, only the red Bok Choy showed anthocyanin accumulation in the leaf veins under LL (Figure 2A), with the anthocyanin content approximately 72% of that under NL (Figure 2A,B). Under LL, the chlorophyll contents of the two *Brassica* cultivars were similar to those found under normal light. However, in the two *Lactuca* species, compared to the green cultivars, the chlorophyll contents in the red cultivars were significantly higher than those in the green cultivars (Figure 2C). Similarly to NL conditions, the *F*_v_/*F*_m_ values were similar across the eight cultivars, with values around 0.8 (Figure 2E).

Similarly to the patterns of activity seen in the four vegetable species under NL conditions, the CLH activity in the green cultivars was also higher than those in the relevant red cultivars under LL. The CLH activity in the green cultivars was in the range of 1–2 μmol Chlide/h/g FW, about 2–3 times higher than that in the red cultivars (Figure 2D). The activity of both the green and red cultivars was lower than under NL, indicating that its dependence on light intensity.

These results indicate that under LL, due to the absence of light stress, the red cultivars did not accumulate anthocyanins in the laminae like the green cultivars. The CLH activities in the green cultivars were lower than those under NL but remained higher than those in the red cultivars.

### 2.3. Responses of the Red-Leaf and Green-Leaf Seedlings Transferred from Low Light to High Light

When seedlings of the four vegetable species grown under LL were transferred to continuous high-light conditions (2500 µmol·m^−2^·s^−1^, HL) for 48 h, the leaves of the red Bok Choy, Ramo-lettuce, and Aspa-lettuce gradually turned from green to red (Figure 3A). After 48 h of HL, the leaves of the red Bok Choy, Ramo-lettuce, and Aspa-lettuce turned dark red, with anthocyanin contents of 0.53 ± 0.05, 0.09 ± 0.01, and 0.10 ± 0.01 mg·g^−1^ FW, respectively, 2.40-, 26.27-, and 5.40-fold higher than under NL, respectively. Red Choy Sum did not show obvious reddening under HL, with 0.0270 ± 0.004 mg·g^−1^ FW after 48 h, only 1.7-fold higher than that under NL, and showed a severe photodamage symptom, such as dehydration and potential necrosis. Under HL, no reddening or anthocyanin accumulation was observed in the four green cultivars (Figure 3B).

Under HL, the chlorophyll content in the green Bok Choy, red/green Ramo-lettuce, and red Aspa-lettuce gradually decreased in the time period from 0 to 24 h, and then increased slightly after 48 h; in red Bok Choy, chlorophyll decreased from 0 to 6 h, then recovered; and in the red/green Choy Sum and green Ramo-lettuce, chlorophyll increased from 0 to 12 h and then decreased (Figure 3C).

The trends of declining *F*_v_/*F*_m_ values in the red/green Bok Choy and Choy Sum plants under HL were similar. The values sharply dropped below 0.2 after 24 h of HL, while the levels in red Choy Sum were significantly lower than that of its green counterpart after 48 h (Figure 3E), dropping below 0.1 and accompanied by leaf dehydration (Figure 3A). The Fv/Fm values of green Ramo-lettuce and green Aspa-lettuce did not decrease significantly, while those of red Ramo-lettuce and red Aspa-lettuce dropped from 7.5 to 0.53 and 0.32, respectively, after 6 h, and then recovered, reaching the values of their counterpart green cultivars after 48 h (Figure 3E).

HL increased CLH activity in all cultivars, with a greater increase seen in green cultivars compared to red. The CLH activity in green Bok Choy was significantly higher than in red Bok Choy after 12 h, reaching about 2.7 times that of the red cultivar at 24 h. Under HL, the CLH activity in green Choy Sum was consistently higher than that in the red cultivar, and about 1.4 times that of the red cultivar at 24 h, after which the increase slowed down. In green Ramo-lettuce, the CLH activity peaked at 24 h, reaching more than 5.96 times that of the red cultivar. In green Aspa-lettuce, the CLH activity increased significantly after 24 h, reaching 11.2 times that of the red cultivar after 48 h.

In summary, when transferred from LL to HL conditions, the seedlings experienced light stress, demonstrating a significant drop in PSII activity (*F*_v_/*F*_m_ values). To alleviate photoinhibition, the red cultivars accumulated more anthocyanins, while the green cultivars were unable to accumulate anthocyanins, but induced higher CLH activity. The green cultivars appeared to have stronger photoprotection even though no anthocyanin was accumulated, showing that CLH induction may be supplementary to the inability to accumulate pigments. The results are in agreement with our previous findings that enhanced CLH activity protects young leaves from light damage [33]. We suggest that photoinhibition can be alleviated through either anthocyanin synthesis or CLH induction under certain conditions, and the two pathways may play complementary roles in photoprotection.

### 2.4. Photosynthetic Apparatus Responses in the Seedlings Transferred from Low Light to High Light

To further understand the photoprotective roles of anthocyanin accumulation and CLH induction, we investigated the photosynthetic apparatus responses in the leaves of the seedlings. Chlorophyll fluorescence images of the seedlings before and after 24 and 48 h of HL exposure revealed an obvious change from blue to brown in most leaves in the four *Brassica* species (Figure 4A), further indicating a significant drop in *F*_v_/*F*_m_, consistent with the values in Figure 3E. In contrast, the leaves of the four *Lactuca* species remained blue, indicating relatively high *F*_v_/*F*_m_, consistent with the values in Figure 3E.

We also compared the light-response curves for the electron transport rates (*ETRs*) and non-photochemical quenching (NPQ) of the seedlings before and after the 24 and 48 h HL treatments (Figure 4B). Before the HL treatment (0 h), the *ETRs* of green Bok Choy and Ramo-lettuce increased more than their red counterparts as the incident light intensity increased from 0 to 1400 µmol·m^−2^·s^−1^, while Choy Sum and Aspa-lettuce showed similar values between the red and green cultivars; *ETR* saturation was observed at 900–1200 µmol·m^−2^·s^−1^ for all the cultivars. After 24 h and 48 h HL treatment, the *ETRs* of the red Bok Choy and red/green Choy Sum were almost undetectable, while the *ETRs* of the green Bok Choy were reduced to a lesser extent, with saturation values decreasing from around 60 (before HL) to 20 and 5 after 24 h and 48 h, respectively. The *ETRs* plotted against the incident light intensity of the 24 h HL-treated red/green *Lactuca* cultivars were similar to the patterns shown before HL treatment, while the rates of Aspa-lettuce were reduced when compared to the rates before HL treatment; after 48 h HL treatment, the ETRs of the four *Lactuca* cultivars were reduced, while higher *ETRs* were detected for the green *Lactuca* cultivars compared to the red ones.

Before HL treatment, the NPQ values of all cultivars showed similar patterns of increasing with the incident light intensity, except for green Ramo-lettuce, where the values increased more than its red counterpart. After 24 and 48 h of high-light treatment, lower NPQ values were found for all four Brassica cultivars compared to the *Lactuca* cultivars, indicating that the seedlings might have suffered light damage severe enough to prevent stimulation of the NPQ protection pathway, particularly for red Choy Sum after 48 h. For the *Lactuca* cultivars, after 24 h HL treatment, higher NPQ induction was found for the red Ramo-lettuce than its green counterpart, while similar patterns were found between the red and green Aspa-lettuce; after 48 h, the two green cultivars demonstrated higher NPQ induction than their red counterparts.

Taken together, after 48 h of HL treatment, the green cultivars showed greater *ETR* induction as the incident light intensity increased compared to their red counterparts, indicating that the green cultivars possessed stronger HL tolerance than the red ones. After 24 h of HL treatment, the green cultivars showed either lower or similar levels of NPQ and higher *F*_v_/*F*_m_ and *ETRs* compared to their red counterparts. This indicates that the green cultivars activated other photoprotective pathways than NPQ, which might be related to the induction of CLH activity.

### 2.5. The Compensatory and Alternative Relationship Between Anthocyanin Accumulation and CLH Induction Is Further Confirmed in the Relevant Arabidopsis Mutant and Overexpressor Lines

To further confirm the alternative relationship between anthocyanin accumulation and CLH in photoprotection, we analyzed these factors in Arabidopsis. *AtPAP1* is a key MYB transcription factor that positively regulates anthocyanin synthesis in Arabidopsis, and its constitutively overexpressing line (*PAP1-D*) accumulates large amounts of anthocyanins. *TT3* and *TT4* encode key enzymes in the anthocyanin biosynthesis pathway: dihydroflavonol-4-reductase (DFR) and chalcone synthase (CHS), respectively. The *tt3 tt4* double mutant shows a deficiency in anthocyanin accumulation. In this study, we treated the Arabidopsis lines *PAP1-D*, *tt3 tt4*, *CLH1*-overexpressing line (*OX-CLH1*), *CLH*-deficient *clh1-1/clh2-2* double mutant, and wild-type (WT) under the high-light conditions described above. Consistent with our previous study [34], after 48 h of HL treatment, the WT leaves turned red and showed anthocyanin accumulation. The *PAP1-D* leaves already accumulated anthocyanins before HL treatment and their leaves turned dark purple after 48 h, containing 330% anthocyanins compared to that of WT. The *tt3 tt4* mutant showed no anthocyanin accumulation and began to exhibit photobleaching after 48 h. The leaves of the *clh1-1/clh2-2* double mutant showed anthocyanin accumulation at 24 h, with the content being 250% that of the WT at 48 h. *OX-CLH1* showed insignificant anthocyanin accumulation, with the content being 70% that of the WT at 48 h (Figure 5B).

We analyzed the expression of the *AtCLH1/2* genes in the *PAP1-D*-overexpressor and *tt3 tt4* mutant lines. The results showed that both *AtCLH1* and *AtCLH2* were upregulated in these lines after 6–12 h of HL treatment. The expression of the two genes in the WT and *tt3 tt4* mutant lines reached the maximum levels at 24 h; *tt3 tt4* showed higher levels than *PAP1-D* and WT (100% and 20% higher than WT at 12 h for *AtCLH1* and *AtCLH2,* respectively), while the expression levels in *PAP1-D* started to decrease after 24 h to only 10% and 30% that of WT, respectively (Figure 5C,D). The results indicate that, in response to HL, the anthocyanin biosynthesis-deficient line *tt3 tt4* induced higher *CLH* gene expression than WT, while in *PAP1-D*, the genes were unable to activate due to strong anthocyanin biosynthesis.

In the WT, *clh1-1/clh2-2* double mutant, and *OX-CLH1* lines, *AtCHS* and *AtDFR* were upregulated by the HL treatment, with the greatest upregulation seen in *clh1-1/clh2-2* and the lowest in *OX-CLH1*. The expression levels peaked at 24 h, where the *AtCHS* and *AtDFR* expression in *clh1-1/clh2-2* was around three-fold higher than in WT and six-fold higher than in *OX-CLH1*. The results indicate that under HL, the *clh1-1/clh2-2* mutant induces strong anthocyanin biosynthesis for protection against light to complement its deficient CLH function, while for the CLH overexpression line, anthocyanin biosynthesis does not appear to be as necessary. These results further confirm the complementary relationship between anthocyanin accumulation and CLH induction that ensures photoprotection in young leaves; when one pathway is deficient, plants compensate by inducing the other pathway to a greater extent.

## 3. Discussion

Anthocyanins have long been considered to serve as screens that reduce the risk of photodamage in vegetative tissues based on their function: reducing light levels incident on photosynthetic mesophyll cells [35]. These pigments are usually accumulated in the epidermal cells of leaves and transiently induced in low-photosynthetic-capacity leaves, such as young leaves or mature leaves facing environmental stresses [36]. However, attempts to determine the photoprotective mechanism of anthocyanins by comparing the photo-responses of red and green leaves from the same genetic background have yielded controversial results. Burger and Edwards [37] reported that reductions in the quantum yield of photosynthetic oxygen evolution caused by experimental exposure to intense visible light did not differ significantly between red- and green-leafed varieties of coleus. In the same species, Logan et al. [38] found that red- and green-leafed coleus varieties exposed to intense white light demonstrated similar reductions in the maximum quantum efficiency of PSII (*F*_v_/*F*_m_). Gould et al. [10] summarized about 50 published reports, with approximately 30% reporting that either the physiological responses of red and green leaves to high light were comparable, or the red leaves performed worse than their green counterparts. When comparing WT and transgenic “Galaxy Gala” apple plants overexpressing *MdMYB10* under high light stress, which dramatically enhanced leaf anthocyanin accumulation, Zhao et al. [39] found that the degree of photoinhibition was comparable between transgenic red leaves and wild-type leaves under field conditions. In the present study, we compared the light responses of both the red- and green-leaved varieties of four leafy vegetable species, each with highly similar genetic backgrounds. We found that the reduction in *F*_v_/*F*_m_ seen in the red- and green-leaved varieties of Bok Choy and Agra-lettuce after high-light treatment were comparable, while the red leaves performed worse than their green counterparts for Choy Sum and Ramo-lettuce (Figure 3E). When their ETRs were plotted against incident light intensity, the red leaves performed either comparably to or worse than their green counterparts for all four species after 24 h or 48 h of HL treatment (Figure 4B). These data are in agreement with the findings of many previous studies, where many red-leaf varieties did not show better high-light tolerance than their green counterparts [10], leaving the photoprotective role of anthocyanins in question.

Regarding the inconsistency of anthocyanin’s role in photoprotection, multiple investigations have been carried out on various aspects of its light response. It was found that light attenuation by anthocyanins could alleviate the degree of photoinhibition under white-, green-, or blue-light stress conditions, while no light attenuation occurred under red-light stress [39]. Wizard Jade coleus, a green-leafed variety, showed significantly higher levels of NPQ than red-leafed Black Dragon over the course of HL exposure, which may have compensated for the absence of visible light screening using an adaxial anthocyanin layer to yield equivalent levels of photoprotection [38]. However, higher levels of NPQ in red leaves than in green were also observed in *Prunus domestica*, in which the green-light attenuation of anthocyanins may impose a limitation on leaf thickness and compromise their protective function [34]. In the present study, similarly low NPQ values were found for all four *Brassica* cultivars after 24 h HL treatment. For the four *Lactuca* cultivars, the green Ramo-lettuce leaves had lower NPQ values than their red counterparts after 24 h HL treatment, while the red/green Aspa-lettuce had similar patterns (Figure 4B). Based on the data showing that the green cultivars had higher *F*_v_/*F*_m_ and ETR values than their red counterparts, we suggest that the lower photodamage after 24 h HL treatment in the green cultivars is not due to higher NPQ protection. One striking difference between the red and green cultivars found in the present study is that all the green cultivars showed higher CLH activity than their red counterparts at all light conditions; markedly higher under both normal light (Figure 1) and when transferred from low- to high-light conditions (Figure 3). Accordingly, we suggest that different plant species activate distinct strategies to cope with the increase in light intensity. We could not deny the photoprotective role of anthocyanins based solely on the fact that red leaves demonstrated no higher light tolerance when compared to the green leaves, since the green leaves induced other photoprotection strategies. Here, we consider that the induction of CLH activity in the green cultivars may compensate for their absence of anthocyanin screening, and the two strategies may play complementary light protection roles at certain developmental stages, e.g., young leaf stage.

CLH was long thought to be the first enzyme in the Chl catabolic pathway, catalyzing the dephytylation of Chl molecules. However, it has since been confirmed that pheophytinases (PPHs) and not CLHs are involved in Chl degradation in leaf senescence, and so the function of CLHs is unclear [40]. Based on the fact that CLHs show high expression and activity in young leaves but not in mature or senescent leaves [31,32], we systematically investigated the function of CLHs in young Arabidopsis leaves in our previous study. We found that Arabidopsis seedlings of *clh1* single and *clh1-1/2-2* double mutants displayed increased photoinhibition after long-term high-light exposure, while seedlings overexpressing CLH1 had enhanced light tolerance compared to the wild type. Recently, CHLOROPHYLL DEPHYTYLASE1 (CLD1) was also identified as a Chl dephytylation enzyme by genetic studies, predominantly expressed in green organs, and its supra-optimal activity in *cld1-1* mutants suffering from heat-dependent photodamage was also demonstrated [41]. It has been suggested that CLD1 is involved in chlorophyll a recycling during its turnover in a steady state; its abnormally high activity leads to the accumulation of phototoxic intermediates such as Chlide a. In our previous study, we found that Arabidopsis CLH1 is associated with PSII-dismantling complexes and responsible for their Chl dephytylation, facilitating core protein D1 degradation in PSII repair, and accordingly plays an important role in photoprotection. These data indicate that Chl catabolism involves different players and is more complicated than previously thought [42].

In the present study, the green cultivars appeared to have stronger photoprotection even though anthocyanin was not accumulated, showing that CLH induction may be supplementary to the inability to accumulate pigments. The results are in agreement with our previous findings that enhanced CLH activity can protect young leaves from light damage [33]. Based on the similar Chl levels before and after HL treatment, we further suggest that the photoprotective role of CLHs is not related to its role in Chl degradation to reduce photosynthetic pigment levels (Figure 3B), but may be via its role in PSII repair as we found for Arabidopsis AtCLH1 [33]. Based on the results of the present study, we propose that photoinhibition in young leaves can be alleviated through either anthocyanin synthesis or CLH induction at certain plant developmental stages, and that the two pathways may play alternative roles in photoprotection to a certain extent. This hypothesis is further supported by our analysis of Arabidopsis lines in the present study, in which the anthocyanin-deficient mutant *tt3 tt4* induced higher *AtCLH1/2* expression than WT and the constitutive accumulation line *PAP1-D*, and almost no anthocyanin accumulation was detected in the *AtCLH1* overexpressor line (Figure 5).

## 4. Materials and Methods

### 4.1. Plant Materials and Growth Conditions

The following vegetable cultivars were used: Bok Choy (*Brassica rapa* var. *chinensis*) (green variety: Siji Shanghai Qing, red variety: Ziguan No. 1); Choy Sum (*Brassica rapa* var. *parachinensis*) (green variety: Youqing 60, red variety: Jiu Yue Xian Hong); Ramo-lettuce (*Lactuca sativa* var. *ramose* Hort) (green variety: Italian Romaine Lettuce, red variety: Guangdong Local Red Romaine Lettuce); and Aspa-lettuce (*Lactuca sativa* var. *asparagina*) (green variety: Siji Asparagus lettuce, red variety: Mianyang Fast-Growing Red Asparagus lettuce). All seeds were obtained from the Vegetable Research Institute of Guangdong Academy of Agricultural Sciences.

Seeds were soaked in water overnight and sown into the soil after swelling. After germination and cotyledon expansion, the seedlings were transplanted individually into pots under conditions of 280 µmol·m^−2^·s^−1^ light intensity (normal light, NL), 20 µmol·m^−2^·s^−1^ (low light, LL), a photoperiod of 12 h day/12 h night, a temperature of 20 °C, and an air relative humidity of 80% in a growth chamber.

Upon development of 2–4 true leaves, the LL-exposed seedlings were transferred to continuous conditions of 2500 µmol·m^−2^·s^−1^ light intensity (high light, HL), 20 °C, and 80% RH. Leaf samples were collected from cultivated NL, LL-, and HL-treated seedlings at 0, 6, 12, 24, and 48 h. All samples were immediately frozen in liquid nitrogen and kept at −80 °C for subsequent analyses.

The *Arabidopsis* (*Arabidopsis thaliana*) lines used were Columbia wild-type (WT), *PAP1*-overexpressing (*PAP1-D*), anthocyanin synthesis gene-deficient double mutant (*tt3 tt4*), CLH-deficient *clh1-1/2-2* double mutants, and *AtCLH1* (*OX-CLH1*) overexpressor. Sterilized seeds were sown on sterile MS medium supplemented with 5% sucrose and 0.8% phytagel and cultivated under 35 µmol·m^−2^·s^−1^ light intensity with a 16 h light/8 h dark photoperiod at 20–22 °C and approximately 80% relative humidity until use. When seedlings developed 4–6 true leaves, they were transferred to continuous HL conditions (680 µmol·m^−2^·s^−1^). Leaves were harvested at 0, 6, 12, 24, and 48 h after high-light exposure, flash-frozen in liquid nitrogen, kept at −80 °C, and used for subsequent experiments.

### 4.2. Anthocyanin Content Determination

Anthocyanin extraction and quantification were performed as described by Fang et al. [43] and Giusti and Wrolstad [44], with minor modifications. A total of 0.1 g of frozen leaf tissue was ground to a fine powder in liquid nitrogen, 1 mL of 0.5% (*v*/*v*) HCl in methanol was added, and the mixture was vortexed and incubated in darkness with periodic shaking until the tissue was completely decolorized. The homogenate was centrifuged (12,000× *g*, 10 min, 4 °C), and the supernatant was collected. The volume of the supernatant was adjusted to 1 mL with extraction buffer.

The anthocyanin content was determined using the pH differential method. Two aliquots (200 μL each) of the extract were separately diluted to 1 mL with 0.4 M KCl-HCl buffer (pH 1.0) and 0.4 M sodium citrate–disodium hydrogen phosphate buffer (pH 4.5). After equilibration for 15 min, the absorbance of each solution was measured at 510 nm and 700 nm against a distilled water blank using a spectrophotometer (Shimadzu UV-2450, Kyoto, Japan). The anthocyanin concentration was calculated as cyanidin-3-glucoside equivalents using the molar extinction coefficient and molecular weight as defined by Giusti and Wrolstad [44].

### 4.3. Chlorophyll Content Determination

Chlorophyll was extracted and quantified according to Schenk et al. [29]. Frozen leaf tissue (0.1 g) was ground in liquid nitrogen. A total of 1 mL of ice-cold extraction buffer (80% (*v*/*v*) acetone containing 10% (*v*/*v*) 0.2 M Tris-HCl, pH 8.0) was added, and the mixture was vortexed and incubated in darkness with periodic shaking until the tissue was completely decolorized. The homogenate was centrifuged (12,000× *g*, 10 min, 4 °C) and the supernatant was collected. The absorbance of the supernatant was measured at 663 nm and 645 nm using 80% acetone as the blank.

### 4.4. Determination of Chlorophyll Fluorescence and Light-Response Curves

Chlorophyll fluorescence parameters were measured using an OS-30P modulated chlorophyll fluorometer (Opti-Sciences, Hudson, NH, USA) after a 20 min dark adaptation period at ambient temperature. For each time point, measurements of the leaves were taken at positions 3–4 of three seedlings for the biological replicates.

Light-response curves for electron transport rate (ETR) and non-photochemical quenching (NPQ) of PSII were generated as described by Lu et al. [45]. Seedlings were exposed to a series of actinic light intensities (0, 81, 145, 186, 281, 335, 461, 701, 926, 1076, and 1251 μmol photons m^−2^ s^−1^). Parameters were recorded on the leaves at positions 3–4 after 10 min of illumination at each intensity. For each plant, measurements were performed in triplicate (technical replicates) at each light level, with three plants measured per time point (biological replicates).

### 4.5. Chlorophyllase Activity Assay

Chlorophyllase (CLH) was extracted as described by Jacob-Wilk et al. [46]. Leaf tissue was homogenized in ice-cold extraction buffer containing 50 mM Na-phosphate pH 7.4, 50 mM KCl, 0.5% Triton X-100, and 5% (*w*/*v*) polyvinylpyrrolidone (PVPP). The mixture was shaken at 300 rpm at 30 °C for 1 h. The homogenate was then centrifuged (15,000× *g*, 20 min, 4 °C), and the supernatant was used as the crude enzyme extract.

Chlorophyll a (Chl a) was isolated from fresh spinach leaves according to Bazzaz and Rebeiz [47] and used as the substrate. The reaction mixture contained 100 μL crude enzyme extract, 400 μL sodium phosphate buffer (0.1 M, pH 7.0), and 100 μL Chl a solubilized in acetone (final acetone concentration < 10% *v*/*v*). The reaction was initiated by adding Chl a and incubated in a shaking water bath at 28 °C for 60 min. The reactions were terminated by transferring 200 μL aliquot to a tube containing 800 μL of acetone/hexane (4:6, *v*/*v*) and vortexed vigorously. The mixture was centrifuged (5000× *g*, 5 min) to separate phases. The concentration of chlorophyllide a (Chlide a) in the acetone phase (lower phase) was determined spectrophotometrically by measuring absorbance at 667 nm. CLH activity was calculated using the extinction coefficient for Chlide a (74.9 mM^−1^ cm^−1^) [48] and expressed as μmol Chlide/h/g Fw.

### 4.6. Gene Expression Analysis with Quantitative Real-Time PCR

Total RNA was extracted from frozen leaf samples using a Plant Total RNA Extraction Kit. Genomic DNA contamination was removed by DNase I treatment. First-strand cDNA was synthesized from 1 μg total RNA using a PrimeScript™ RT reagent kit according to the manufacturer’s instructions.

Quantitative real-time PCR (qRT-PCR) was performed using a CFX96 Real-Time PCR Detection System (Bio-Rad, Hercules, CA, USA) with gene-specific primers. All the accession numbers and primers of the related genes are listed in Table A1. Relative transcript levels were calculated using the 2^−ΔCt^ method, with *AtActin7* (AT5G09810) as the internal reference gene. Three technical replicates were performed for each cDNA sample, and three independent biological replicates (plants) were analyzed per genotype and time point. Data are presented as mean relative transcript level ± SE (*n* = 9; 3 biological × 3 technical replicates).

### 4.7. Statistics Analysis

The values are the mean ± standard error (SE; *n* = 3, except that it is indicated). Data processing and statistical analyses were performed using IBM SPSS Statistics 20 software. One-way analysis of variance (ANOVA) with Tukey’s method was applied for the red and green leaf samples of the same species grown in low and normal light, and for all the red and green leaf samples along the time course of high light treatments. All graphical representations were generated with OriginPro 2021 software.

## 5. Conclusions

Our comparison of the light response and the relationship between anthocyanin accumulation and CLH activation in four leafy vegetable species, each with both red- and green-leaved varieties, showed that the red-leaf cultivars had either comparable or worse high-light tolerance than their green counterparts. We found that the greater induction of CLH activity in the green cultivars may compensate for their absence of anthocyanin screening and even afford better photoprotection, and that the two strategies may play complementary light protection roles.

## Figures and Tables

**Figure 1 plants-14-02950-f001:**
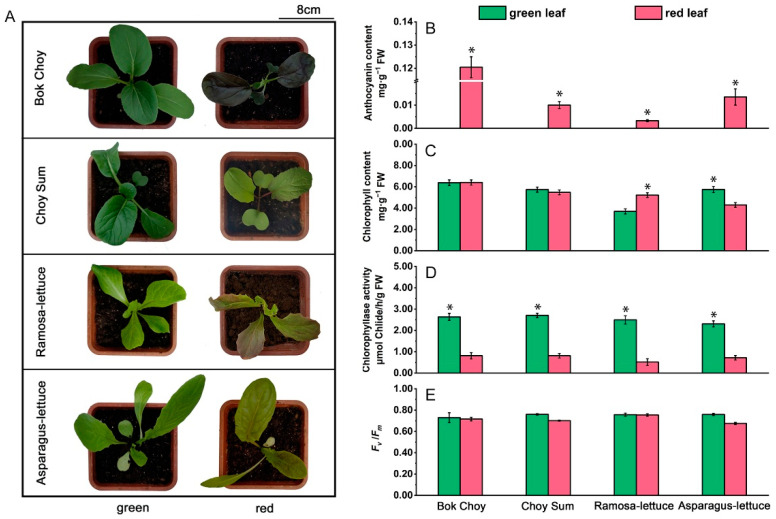
Anthocyanin content and CLH activity in the leaves of the eight red/green-leafed vegetable seedlings grown under normal-light conditions. (**A**) The appearance of the seedlings grown under normal light (NL). Seedlings with 4 true leaves from 4 leafy vegetable species (Bok Choy (*Brassica rapa* var. chinensis), Choy Sum (*Brassica rapa* var. Parachinensis), Ramosa lettuce (*Lactuca sativa* var. ramosa), and Asparagus lettuce (*Lactuca sativa* var. asparagine)), each with both red and green cultivars (a total of 8 cultivars), were grown under NL (280 µmol·m^−2^·s^−1^). (**B**) The anthocyanin content of leaves at positions 3–4 (low to high) in the seedlings described in (**A**). (**C**) The chlorophyll content of the leaves described in (**B**). (**D**) Chlorophyllase (CLH) activity in the leaves described in (**B**). (**E**) *F*_v_/*F*_m_ values in the leaves described in (**B**). Error bars in (**B**–**E**) indicate standard error (SE). Significant differences (*p* < 0.01) between the red- and green-leafed cultivars of the same species were labeled with asterisks for the greater values.

**Figure 2 plants-14-02950-f002:**
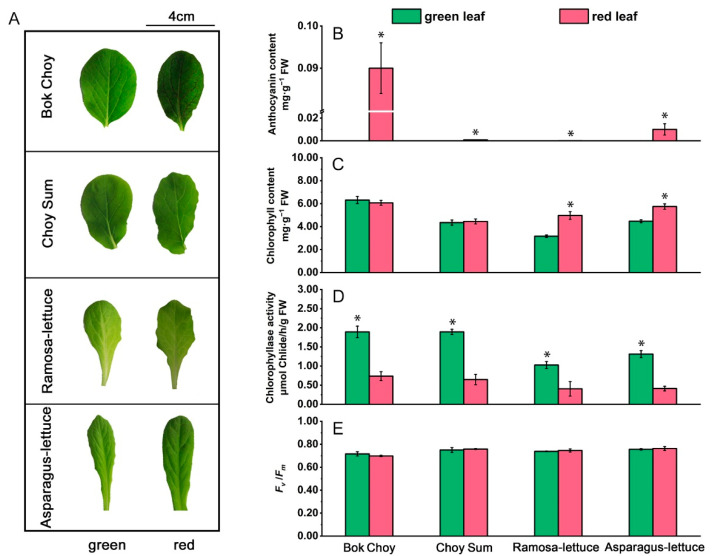
Anthocyanin content and CLH activity in the leaves of the eight red/green-leafed vegetable seedlings grown under low-light conditions. (**A**) The appearance of the 4th leaf from each of the eight red/green-leafed vegetable seedlings grown under low light (LL, 20 µmol·m^−2^·s^−1^), as described in Figure 1A. (**B**) The anthocyanin content in the leaves at positions 3–4 (low to high) from the seedlings described in (**A**). (**C**) The chlorophyll content in the leaves described in (**B**). (**D**) Chlorophyllase (CLH) activity in the leaves described in (**B**). (**E**) *F*_v_/*F*_m_ values of the leaves described in (**B**). Error bars in (**B**–**E**) indicate standard error (SE). Significant differences (*p* < 0.01) between the red- and green-leafed cultivars of the same species were labeled with asterisks for the greater values.

**Figure 3 plants-14-02950-f003:**
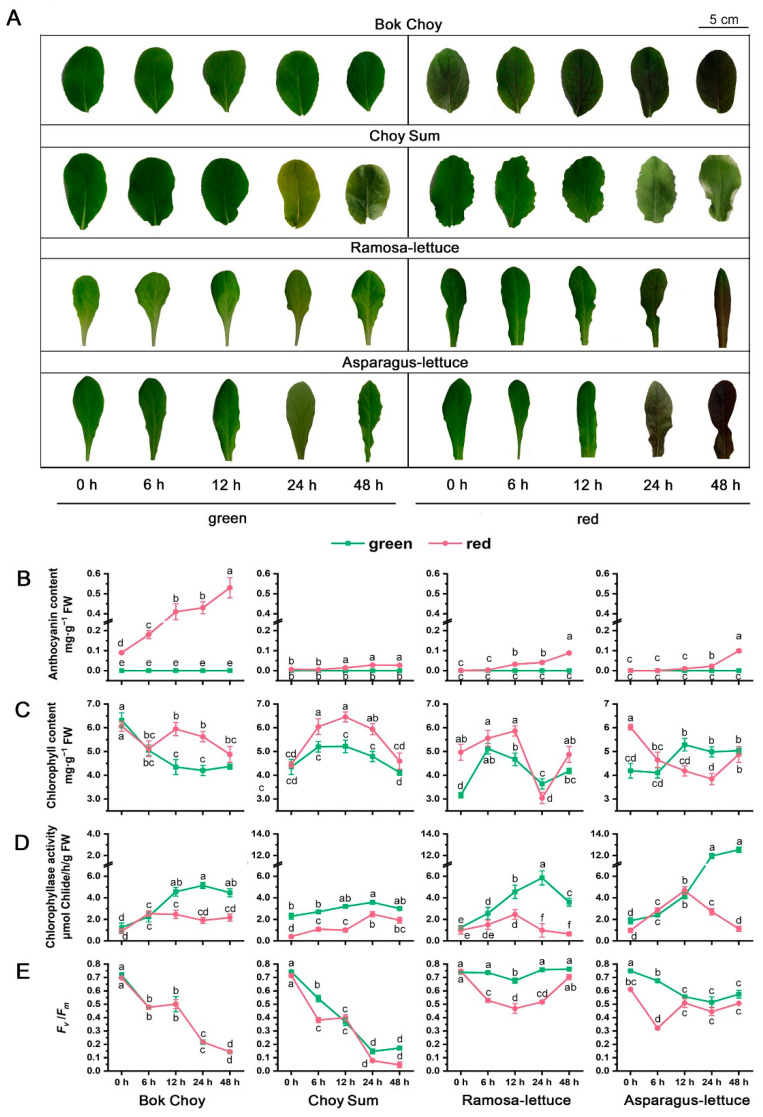
Anthocyanin content and CLH activity in the leaves of the eight red/green-leafed vegetable seedlings exposed to high light. (**A**) Appearance of the eight red/green-leafed vegetable leaves transferred from low-light conditions (LL, 20 µmol·m^−2^·s^−1^) and exposed to high light (HL, 2500 µmol·m^−2^·s^−1^) for 48 h. The seedlings described in Figure 1A were grown under LL until 4 true leaves were seen, and then exposed to HL. The figure shows the 4th leaves of each seedling at 0, 6, 12, 24, and 48 h under HL conditions. (**B**) The anthocyanin content in leaves 3–4 (low to high) of the seedlings described in (**A**). (**C**) The chlorophyll content in the leaves described in (**B**). (**D**) Chlorophyllase (CLH) activity in the leaves described in (**B**). (**E**) *F*_v_/*F*_m_ values in the leaves described in (**B**). Error bars in (**B**–**E**) indicate standard error (SE). Significant differences (*p* < 0.01) between the samples of the red- and green-leafed cultivars of the same species along the time course of HL treatment were labeled by different letters, a, b, c, d, and e.

**Figure 4 plants-14-02950-f004:**
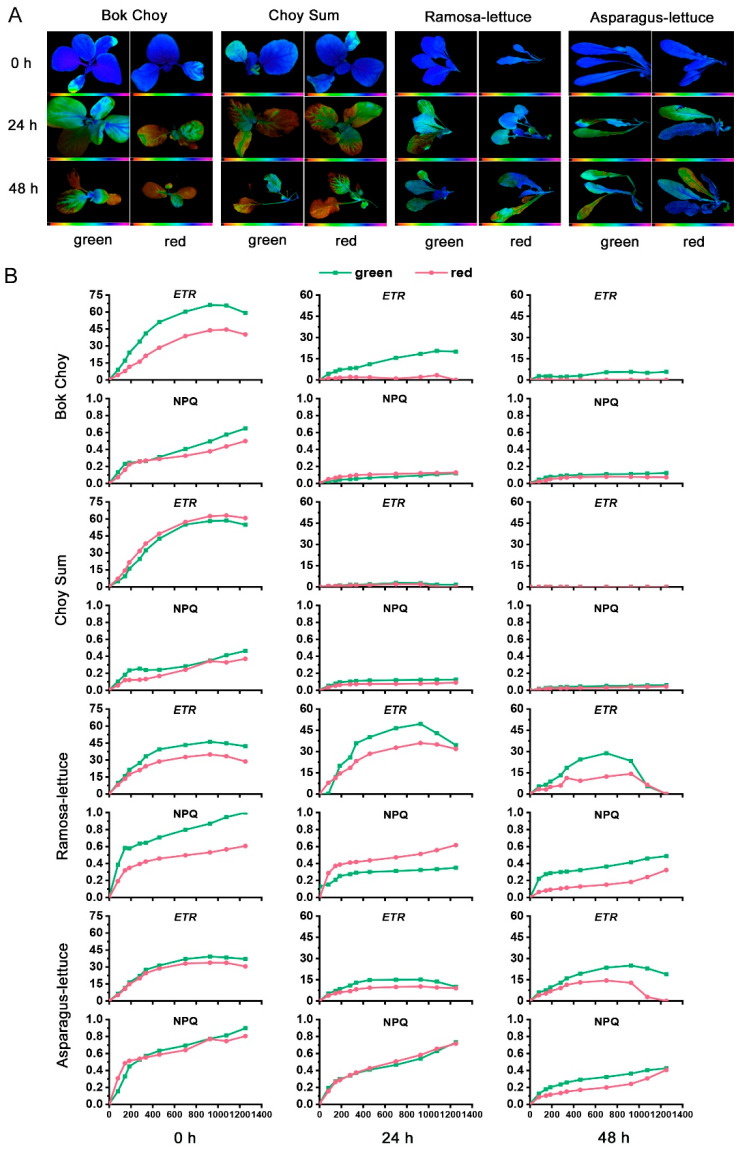
False-color images and light-response curves for the leaves of the eight red/green-leafed vegetable seedlings exposed to high-light conditions. (**A**) False-color images representing the maximal photochemical efficiency of photosystem II (PSII; *F*_v_/*F*_m_) in the seedlings described in Figure 3A before (0 h) and after 24 and 48 h HL treatment. The false-color scale ranges from black (0) via red, orange, yellow, green, blue, and violet to purple (1), as indicated below the false-color images. (**B**) Light-response curves of the electron transport rates (ETRs) and non-photochemical quenching (NPQ) of PSII in the 4th leaves of the seedlings, as described in (**A**), before and after 24 and 48 h HL treatment. Measurements were performed at light intensities of 0, 81, 145, 186, 281, 335, 461, 701, 926, 1076, and 1251 μmol photons·m^−2^·s^−1^.

**Figure 5 plants-14-02950-f005:**
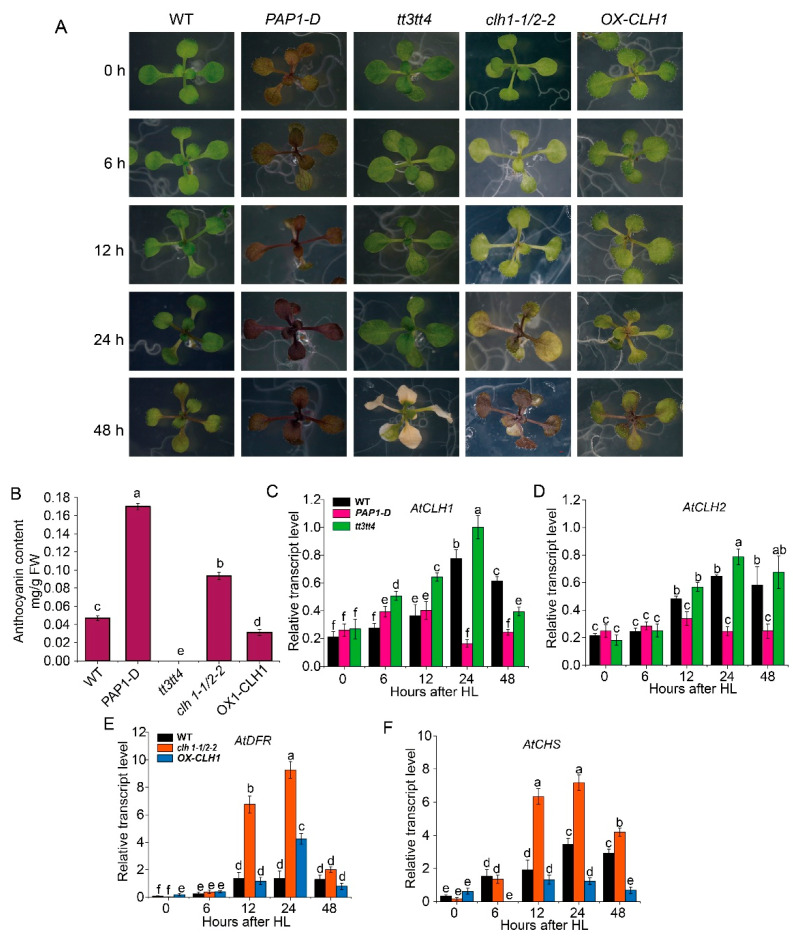
Anthocyanin accumulation and *AtCLH1/2* gene expression in the high-light-treated seedlings of Arabidopsis lines in which anthocyanin is constitutively or deficiently accumulated, and in *CLH*-null or -overexpressing lines. (**A**) Seedlings of Arabidopsis lines in which anthocyanin is constitutively (*PAP1-D*) or deficiently (*tt3 tt4*) accumulated; CLH-null (*clh1-1/2-2*) and -overexpressing (*OX-CLH1*) lines; and wild-type (WT) lines exposed to HL. The seedlings were germinated and grown under low-light conditions (35 μmol photons m^−2^·s^−1^; LL) for 16 d until the appearance of 4 true leaves. Then, the seedlings were transferred and exposed to HL (680 μmol photons m^−2^·s^−1^) for 48 h. Similar seedling images were obtained from multiple independent replicates, and representative pictures of plants at different time points during HL exposure are shown. (**B**) Anthocyanin contents in the 3rd and 4th leaves of the lines described in (**A**) after 48 h HL exposure. (**C**,**D**) Relative expression levels of *AtCHL1* (**C**) and *AtCHL2* (**D**) in the 3rd and 4th leaves of WT, PAP1-D, and *tt3 tt4* seedlings under HL conditions. (**E**,**F**) Relative expression levels of *AtDFR* (**E**) and *AtCHS* (**F**) in the 3rd and 4th leaves of WT, *clh1-1/2-2*, and *OX-CLH1* seedlings under HL conditions. Error bars in (**C**–**F**) indicate SE; one-way analysis of variance (ANOVA) with Tukey’s method was applied to show the significant differences at *p* < 0.01 in pairwise comparison and classified with the letters a, b, c, d, e, or f.

## Data Availability

The original contributions presented in this study are included in the article. Further inquiries can be directed to the corresponding author.

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
