# Peer review of "Comparing Four Red/Green-Leafed Vegetables Reveals the Complementary Photoprotective Roles of Anthocyanin Accumulation and Chlorophyllase"

_plants, 2025, doi:10.3390/plants14192950_

Round 1
Reviewer 1 Report
Comments and Suggestions for Authors
Abstract: please explain the CHL acronym
Line 53: phosphorous with capital “P”
Figure 3: when you graphically represent time series for comparison between two groups (i.e., green and red) is better to do it with lines instead of bars, like in Figure 4. In such way the time evolution is more concluding.
Figures 3 and 4: For a better result of visual inspection of the figures, probably should be represented in Landscape mode on all the page.
Figure 5: The diagrams are too small and the statistical significance letters can not be read easily. Please try to enlarge them.
Statistical analysis:
In Figures 1- 5 you inserted the statistical significance from a statistical test, but only in Figure 5 caption you name the statistical test. Is compulsatory to mention the statistical test and the algorithm you used (i.e., between which factor levels) in the 4.7 subsection. Also, the factors you considered, must be described properly. More precisely, in Figure 3 you have three factors the Cultivar, Leaves Colour and Time; please describe more carefully what you comparisons you made that generated the statistical significances described with stars.
Because you have time series involved in the analysis, I expected results from correlations between the red and green responses and some function fittings…
Author Response
Comments and Suggestions by reviewer 1
Abstract: please explain the CHL acronym
Response: we have added the CLH acronym following chlorophyllases in line 19.
Line 53: phosphorous with capital “P”
Response: we have revised and started with the capital.
Figure 3: when you graphically represent time series for comparison between two groups (i.e., green and red) is better to do it with lines instead of bars, like in Figure 4. In such way the time evolution is more concluding.
Response: we have revised the Figure 3B and represented the data with lines.
Figures 3 and 4: For a better result of visual ins pection of the figures, probably should be represented in Landscape mode on all the page.
Response: we have remodeled Figure 3-4 and showed the figures in Landscape mode.
Figure 5: The diagrams are too small and the statistical significance letters can not be read easily. Please try to enlarge them.
Response: we have enlarged the size of the letters in Figure 5.
Statistical analysis:
In Figures 1- 5 you inserted the statistical significance from a statistical test, but only in Figure 5 caption you name the statistical test. Is compulsatory to mention the statistical test and the algorithm you used (i.e., between which factor levels) in the 4.7 subsection. Also, the factors you considered, must be described properly. More precisely, in Figure 3 you have three factors the Cultivar, Leaves Colour and Time; please describe more carefully what you comparisons you made that generated the statistical significances described with stars.
Because you have time series involved in the analysis, I expected results from correlations between the red and green responses and some function fittings
Response: Thanks for your comment. We have a description regarding the significant label stars in the caption of figure 1. In figure 2-3, we described the meaning of the stars as “as described in figure 1“. It might be fine in figure 2. while for figure 3B, you are right that we have more factors, such as species, red/green leafed, and time course. The description as in figure 1 is not clear for this figure.
In the revised version, we have added the individual description for each figure. Also, we have added the statistical analysis method in the 4.7 subsection.
Reviewer 2 Report
Comments and Suggestions for Authors
The ms “Comparison of 4 red/green leafed vegetables reveals the complementary photoprotection roles of anthocyanin accumulation and chlorophyllase” significantly contributes to the topic by introducing chlorophyllase (CLH) activity as a novel photoprotective mechanism. The experimental evidence provided across four leafy vegetable species and Arabidopsis mutants shows that green cultivars exhibit higher CLH activity, especially under high light, while red cultivars accumulate anthocyanins, suggesting a complementary relationship where plants utilize either anthocyanin synthesis or CLH induction for photoprotection, potentially through CLH's role in Photosystem II repair.
ABS – line 27
The sentence in the ABS “AtCLH1 over-expressor and clh1-1/2-2 mutant activated no and more anthocyanin than wild-type, respectively” is grammatically a bit awkward and has a small factual inaccuracy when compared to the more detailed data in the body of the text. It would have been more correct to use "accumulated" or "produced", since anthocyanins are molecules that accumulate in tissues. Moreover, the ABS states that there was “no” anthocyanin activation. However, the results section specifies that the OX-CLH1 line (which overexpresses AtCLH1) showed “insignificant” anthocyanin accumulation, quantified as 70% of the wild-type (WT) content after 48 hours of intense light treatment. Therefore, although accumulation is reduced compared to the WT, it is not zero. This is an oversimplification for the ABS.
Results 2.2 – line 172
The statement that green cultivars do not accumulate anthocyanins appears pleonastic because, by definition within the context of the study, green cultivars are characterized as being "anthocyanin-deficient" or "deficient in anthocyanin accumulation".
Some typos in the text in the attached file

Author Response
Comments and Suggestions by reviewer 2
The ms “Comparison of 4 red/green leafed vegetables reveals the complementary photoprotection roles of anthocyanin accumulation and chlorophyllase” significantly contributes to the topic by introducing chlorophyllase (CLH) activity as a novel photoprotective mechanism. The experimental evidence provided across four leafy vegetable species and Arabidopsis mutants shows that green cultivars exhibit higher CLH activity, especially under high light, while red cultivars accumulate anthocyanins, suggesting a complementary relationship where plants utilize either anthocyanin synthesis or CLH induction for photoprotection, potentially through CLH's role in Photosystem II repair.
ABS – line 27
The sentence in the ABS “AtCLH1 over-expressor and clh1-1/2-2 mutant activated no and more anthocyanin than wild-type, respectively” is grammatically a bit awkward and has a small factual inaccuracy when compared to the more detailed data in the body of the text. It would have been more correct to use "accumulated" or "produced", since anthocyanins are molecules that accumulate in tissues. Moreover, the ABS states that there was “no” anthocyanin activation. However, the results section specifies that the OX-CLH1 line (which overexpresses AtCLH1) showed “insignificant” anthocyanin accumulation, quantified as 70% of the wild-type (WT) content after 48 hours of intense light treatment. Therefore, although accumulation is reduced compared to the WT, it is not zero. This is an oversimplification for the ABS.
Response: Thanks for your comment. We have revised the abstract sentence to “AtCLH1 over-expressor and clh1-1/2-2 mutant accumulated less and more anthocyanin than the wild-type, respectively”.
Results 2.2 – line 172
The statement that green cultivars do not accumulate anthocyanins appears pleonastic because, by definition within the context of the study, green cultivars are characterized as being "anthocyanin-deficient" or "deficient in anthocyanin accumulation".
Response: Thanks for your comment. You are right that we described “the green cultivars do not accumulate anthocyanin” appears pleonastic. While we consider that, even though the cultivars were reported to be “anthocyanin-deficient”, it is necessary to confirm their phenotype in our study.
Reviewer 3 Report
Comments and Suggestions for Authors
This manuscript investigates the relationship between two potential photoprotective mechanisms, anthocyanin accumulation and chlorophyllase (CLH) activity, in plants under varying light conditions. The authors compared red-leafed (anthocyanin-rich) and green-leafed (anthocyanin-deficient) cultivars of four different leafy vegetables (Brassica rapa and Lactuca sativa). The study finds that while red-leafed cultivars accumulate anthocyanins, their green-leafed counterparts consistently exhibit significantly higher CLH activity, particularly when exposed to high light stress. In high light experiments, the green cultivars showed comparable or even better phototolerance than the red cultivars, despite lacking anthocyanins. The authors support these findings with experiments on Arabidopsis mutants, showing that anthocyanin-deficient mutants upregulate CLH genes, while CLH-deficient mutants upregulate anthocyanin synthesis genes. The authors conclude that CLH activity serves as a complementary photoprotective strategy that compensates for the absence of anthocyanins in green leaves.
Major Issues
- The study presents a strong correlation between the absence of anthocyanins and increased CLH activity, but the conclusion of a direct compensatory or complementary role needs to be better substantiated. The current data does not fully establish a causal link showing that the deficiency in one pathway directly triggers the upregulation of the other. For example, it is possible that the genetic selection for the green-leaf phenotype is linked to other traits, including higher basal and inducible CLH activity, which co-evolved as a separate high-light tolerance mechanism rather than a direct compensation. While the Arabidopsis experiments (Figure 5) provide stronger evidence for this interplay, the manuscript would benefit from a more cautious interpretation in the abstract and discussion, or from providing more direct evidence of the regulatory crosstalk between these two distinct metabolic pathways in the vegetable species studied.
- A significant discrepancy exists between the light intensities used for the vegetable experiments and the Arabidopsis experiments, which complicates the direct comparison and generalization of the results. The vegetables were exposed to extremely high light (HL) at 4000 µmol·m⁻²·s⁻¹ (Page 7, line 228), whereas Arabidopsis was exposed to a much lower HL of 680 µmol·m⁻²·s⁻¹ (Page 11, line 329). The intensity used for the vegetables is severe enough to cause visible cell death and dehydration, particularly in the Brassica species (Figure 3A, Page 7, lines 201-202), which may mask or override more nuanced photoprotective responses. The authors should provide a rationale for using such different and extreme light conditions and discuss how this might affect the interpretation of the results, especially when drawing parallels between the two experimental systems.
- The proposed mechanism for CLH-mediated photoprotection is based entirely on the authors' previous work (Tian et al., 2021) and is not directly tested in the vegetable species examined in this study. The manuscript concludes that the role of CLH is not to reduce overall chlorophyll levels but to facilitate PSII repair (Page 13, lines 420-423). This is a critical point that relies on inference rather than direct evidence from the current experiments. To strengthen this claim, the authors could include data on the levels of key PSII components, such as the D1 protein, or other markers of PSII turnover and repair in the red and green vegetable cultivars under high light. Without such data, the mechanistic link between the observed high CLH activity and enhanced phototolerance in the green cultivars remains speculative.
Minor Issues
- Several figures and their descriptions could be improved for clarity. For instance, in Figure 3, the images of the ChoySum leaves after 48 hours of high light show severe damage that appears to be more than just photoinhibition, including dehydration and potential necrosis. This should be more explicitly described in the results text and figure legend to accurately reflect the plants' condition.
- The definition of "normal light" (NL) as 1000 µmol·m⁻²·s⁻¹ (Page 4, line 115) is unusually high for typical growth chamber conditions and could be considered a moderate-to-high light stress level for many plants. This high baseline may influence the plants' physiological state even before the high-light experiment begins. It would be beneficial for the authors to justify this choice of light intensity for NL or acknowledge that the plants were already acclimated to high-light conditions, which could impact their subsequent response to the even more extreme 4000 µmol·m⁻²·s⁻¹ treatment.
- The English language throughout the manuscript, while generally understandable, could be improved for clarity, precision, and conciseness to meet the standards of a scientific publication. There are instances of awkward phrasing, incorrect word choices, and overly complex sentence structures. For example, the sentence "Anthocyanin photoprotection role in leaves is questionable, since some red leaves (anthocyanin-rich) perform no better high-light tolerance than green leaves (anthocyanin-deficient)" (Page 1, lines 13-15) could be rephrased more formally as, "The photoprotective role of anthocyanins in leaves is debated, as some anthocyanin-rich red leaves do not exhibit greater high-light tolerance than their anthocyanin-deficient green counterparts." Another example is on Page 12, line 370, where "mutil-aspects" should be corrected to "multi-aspects" or "multiple aspects." A thorough review by a native English speaker or a professional editing service is recommended to improve the overall readability of the manuscript.
Author Response
Comments and Suggestions by reviewer 3
Comments and Suggestions for Authors
This manuscript investigates the relationship between two potential photoprotective mechanisms, anthocyanin accumulation and chlorophyllase (CLH) activity, in plants under varying light conditions. The authors compared red-leafed (anthocyanin-rich) and green-leafed (anthocyanin-deficient) cultivars of four different leafy vegetables (Brassica rapa and Lactuca sativa). The study finds that while red-leafed cultivars accumulate anthocyanins, their green-leafed counterparts consistently exhibit significantly higher CLH activity, particularly when exposed to high light stress. In high light experiments, the green cultivars showed comparable or even better phototolerance than the red cultivars, despite lacking anthocyanins. The authors support these findings with experiments on Arabidopsis mutants, showing that anthocyanin-deficient mutants upregulate CLH genes, while CLH-deficient mutants upregulate anthocyanin synthesis genes. The authors conclude that CLH activity serves as a complementary photoprotective strategy that compensates for the absence of anthocyanins in green leaves.
Major Issues
- The study presents a strong correlation between the absence of anthocyanins and increased CLH activity, but the conclusion of a direct compensatory or complementary role needs to be better substantiated. The current data does not fully establish a causal link showing that the deficiency in one pathway directly triggers the upregulation of the other. For example, it is possible that the genetic selection for the green-leaf phenotype is linked to other traits, including higher basal and inducible CLH activity, which co-evolved as a separate high-light tolerance mechanism rather than a direct compensation. While the Arabidopsis experiments (Figure 5) provide stronger evidence for this interplay, the manuscript would benefit from a more cautious interpretation in the abstract and discussion, or from providing more direct evidence of the regulatory crosstalk between these two distinct metabolic pathways in the vegetable species studied.
Response: Thanks for your comment. We agree with your comment that the “the conclusion of a direct compensatory or complementary role needs to be better substantiated”. Our study is for the first time to investigate the relation between the two strategies. And based on the data in our study we suggest a complementary relation between the photoprotection roles of anthocyanin accumulation and chlorophyllases. We believe that even though more substantiated evidences are required to demonstrate the relation, it is an interesting and novel findings, which are good for explaining the question whether anthocyanins have the photoprotective roles.
You mentioned that it is possible as a result of genetic selection, rather than a direct compensation relation. Regarding to this issue, we consider that since we have demonstrated that chlorophyllases have photoprotection role in our previous study (Tian et al., 2021), and anthocyanins have long been believed to possess photoprotection role. If the lack of anthocyanins (green) and the induction of CLH activity are the co-evolved high light tolerance mechanism being selected, this may be a good evidence to support the hypothesis that either pathway is necessary for photoprotection at certain developmental stages of plants.
Accordingly, we revised some sentences in the abstract (line 32-34) and discussion (line 439), so as to make cautious interpretation our results.
- A significant discrepancy exists between the light intensities used for the vegetable experiments and the Arabidopsis experiments, which complicates the direct comparison and generalization of the results. The vegetables were exposed to extremely high light (HL) at 4000 µmol·m⁻²·s⁻¹ (Page 7, line 228), whereas Arabidopsis was exposed to a much lower HL of 680 µmol·m⁻²·s⁻¹ (Page 11, line 329). The intensity used for the vegetables is severe enough to cause visible cell death and dehydration, particularly in the Brassica species (Figure 3A, Page 7, lines 201-202), which may mask or override more nuanced photoprotective responses. The authors should provide a rationale for using such different and extreme light conditions and discuss how this might affect the interpretation of the results, especially when drawing parallels between the two experimental systems.
Response: Thanks for your comment. Regarding to the light intensity issue, we checked our data carefully. You mentioned that we used different light intensity treatment for the vegetable and Arabidopsis seedlings. This comment helped us to find out our problem. We‘re-measured the light intensity of the same condition again. This time we found that the light intensity measurement for the Arabidopsis seedlings was as the former, indicating that the light intensity for the Arabidopsis is out of problem. While we found that the former measurement for the vegetables was problematic, due to probability that the instrument we used at that time was not stable. After we measured the intensity with the same conditions we used for the treatment for the vegetables. We got the correct intensities of normal light (NL) and high light (HL) as 280 and 2500 µmol·m-2·s-1 respectively. We have revised the wrong intensities to the correct ones. We are extremely sorry that we presented wrong light intensities in our former version of manuscript. In addition, since Arabidopsis seedlings are cultivated in sterilized agar medium in sealed plates, we used lower (680 µmol·m-2·s-1) light intensity as the high light condition for the Arabidopsis seedlings.
The proposed mechanism for CLH-mediated photoprotection is based entirely on the authors' previous work (Tian et al., 2021) and is not directly tested in the vegetable species examined in this study. The manuscript concludes that the role of CLH is not to reduce overall chlorophyll levels but to facilitate PSII repair (Page 13, lines 420-423). This is a critical point that relies on inference rather than direct evidence from the current experiments. To strengthen this claim, the authors could include data on the levels of key PSII components, such as the D1 protein, or other markers of PSII turnover and repair in the red and green vegetable cultivars under high light. Without such data, the mechanistic link between the observed high CLH activity and enhanced phototolerance in the green cultivars remains speculative.
Response: Thanks for your comment. Based on the findings of the present study that no linear changes between the chlorophyll contents and of CLH activity, we suggest that the role of CLH is not to reduce overall chlorophyll levels, but might play roles to facilitate PSII repair as the roles we found for Arabidopsis CLH1/2 (Tian et al.,2021). You are right that the speculation of their roles in PSII repair is based on our previous findings in Arabidopsis. To confirm this suggestion, evidences collected from the vegetables are required. Since the aim for the present study is to investigate the relation between anthocyanin accumulation and CLH activity, it will be an important issue for the future work. Nevertheless, we revised the sentence for this claim with cautious description (line 470-472).
Minor Issues
- Several figures and their descriptions could be improved for clarity. For instance, in Figure 3, the images of the ChoySum leaves after 48 hours of high light show severe damage that appears to be more than just photoinhibition, including dehydration and potential necrosis. This should be more explicitly described in the results text and figure legend to accurately reflect the plants' condition.
Response: Thanks for your comment. You are right. Among the 4 species, ChoySum is the ones with the least resistance to high light stress. In our former version, we entirely focused on the comparison between the responses of green and red leaves, and missed some important points, such as some different responses between the species, e.g. the severe damage of ChoySum under high light. We have added the description in line 213.
- The definition of "normal light" (NL) as 1000 µmol·m⁻²·s⁻¹ (Page 4, line 115) is unusually high for typical growth chamber conditions and could be considered a moderate-to-high light stress level for many plants. This high baseline may influence the plants' physiological state even before the high-light experiment begins. It would be beneficial for the authors to justify this choice of light intensity for NL or acknowledge that the plants were already acclimated to high-light conditions, which could impact their subsequent response to the even more extreme 4000 µmol·m⁻²·s⁻¹ treatment.
Response: Thanks for your comment. As we mentioned above, we have found out that we had some problems in the intensity measurement for the vegetable experiments. We now have measured the intensities again and revised the light intensity as normal light (NL, 280 µmol·m-2·s-1) and high light (HL, 2500 µmol·m-2·s-1). In addition, to get obvious phenotypes of photoinhibtion or photodamage, we acclimated the seedlings under low light prior to high light treatment.
Comments on the Quality of English Language
- The English language throughout the manuscript, while generally understandable, could be improved for clarity, precision, and conciseness to meet the standards of a scientific publication. There are instances of awkward phrasing, incorrect word choices, and overly complex sentence structures. For example, the sentence "Anthocyanin photoprotection role in leaves is questionable, since some red leaves (anthocyanin-rich) perform no better high-light tolerance than green leaves (anthocyanin-deficient)" (Page 1, lines 13-15) could be rephrased more formally as, "The photoprotective role of anthocyanins in leaves is debated, as some anthocyanin-rich red leaves do not exhibit greater high-light tolerance than their anthocyanin-deficient green counterparts." Another example is on Page 12, line 370, where "mutil-aspects" should be corrected to "multi-aspects" or "multiple aspects." A thorough review by a native English speaker or a professional editing service is recommended to improve the overall readability of the manuscript.
Response: Thanks a lot for your kind help. We accept your suggestion of the sentence rephasing for the sentence in the abstract and revised the “mutil” wrong word. In addition, we have sent our manuscript to MDPI author servers for English editing. We find the English has been improved a lot after the editing.